# Quantiferon Monitor Testing Sheds Light on Immune System Disparities between Multiple Sclerosis Patients and Healthy Individuals

**DOI:** 10.3390/ijms25042179

**Published:** 2024-02-11

**Authors:** Ilona Součková, Ondřej Souček, Jan Krejsek, Oldřich Vyšata, David Matyáš, Marek Peterka, Michal Novotný, Pavel Kunc, Zbyšek Pavelek

**Affiliations:** 1Department of Clinical Immunology and Allergology, University Hospital Hradec Kralove, 50005 Hradec Kralove, Czech Republic; 2Faculty of Medicine in Hradec Králové, Charles University, 50003 Hradec Kralove, Czech Republiczbysek.pavelek@fnhk.cz (Z.P.); 3Department of Neurology, University Hospital Hradec Kralove, 50005 Hradec Kralove, Czech Republic

**Keywords:** multiple sclerosis, immunity, fingolimod

## Abstract

The aim of this study was to conduct QuantiFERON Monitor (QFM) testing in patients with multiple sclerosis (MS), which is used to monitor the state of the immune system through the non-specific stimulation of leukocytes followed by determining the level of interferon-gamma (IFN-γ) released from activated cells. Additionally, we tested the level of selected cytokines (IFN-α, IFN-γ, IL-1α, IL-1β, IL-1ra, IL-2, IL-3, IL-4, IL-6, IL-7, IL-10, IL-15, IL-33, VEGF) from stimulated blood samples to further understand the immune response. This study builds upon a previously published study, utilizing activated serum samples that were initially used for IFN-γ determination. However, our current focus shifts from IFN-γ to exploring other cytokines that could provide further insights into the immune response. A screening was conducted using Luminex technology, which yielded promising results. These results were then further elaborated upon using ELISA to provide a more detailed understanding of the cytokine profiles involved. This study, conducted from August 2019 to June 2023, included 280 participants: 98 RRMS patients treated with fingolimod (fMS), 96 untreated patients with progressive MS (pMS), and 86 healthy controls (HC). Our results include Violin plots showing elevated IL-1α in pMS and fMS. Statistical analysis indicated significant differences in the interleukin levels between groups, with IL-1ra and age as key predictors in differentiating HC from pMS and IL-1ra, IL-1α, age, and EDSS in distinguishing pMS from fMS. These findings suggest cytokines’ potential as biomarkers in MS progression and treatment response.

## 1. Introduction

Multiple sclerosis (MS) is a chronic autoimmune disorder characterized by the immune system’s aberrant attack on the central nervous system (CNS), leading to demyelination and neurodegeneration [1]. The pathophysiology of MS involves complex interactions between immune cells, CNS resident cells, and various cytokines and chemokines, resulting in inflammation and neuronal damage.

MS is currently recognized as abnormally polarized Th1 and Th2 T cell subset-driven immunopathology with the substantial contribution of B cells. This paradigm is evidenced by numerous studies exploiting both animal models EAE and clinical studies focusing on people with MS. However, the most decisive are the results of many large clinical trials showing the positive impact of drugs with different mechanisms of action on inflammatory response in the natural course of MS patients. The spectrum of already approved disease-modifying drugs (such as DMTs) is very broad. From the perspective of molecular structure, they comprise small molecules up to complex biological therapeutics. Mechanisms of action are also very different when compared with various DMTs. Some DMTs are recognized as immunomodulatory (recombinant interferons β, glatiramer), and some have such complex actions on the immune system of MS patients that they are denoted as immunoreconstitution therapies (cladribine, antiCD20 biologics, antiCD52 biologic), some biological therapies can block the entry of lymphocytes through the blood–brain barrier into the brain (natalizumab), and some small molecules block the aggress of especially T and B cells from secondary lymphoid organs (sphingosine phosphate receptors modulators) [1,2].

Over the past few decades, the therapeutic landscape of MS has evolved significantly, with the introduction of various disease-modifying therapies (DMTs) aimed at modulating or suppressing immune response. Fingolimod (FTY720), approved by the FDA in 2010, emerged as the first oral DMT for relapsing forms of MS, marking a significant milestone in MS treatment [3]. Fingolimod is a sphingosine 1-phosphate (S1P) receptor modulator. It acts by binding to S1P receptors on lymphocytes, leading to their sequestration in lymph nodes and thereby reducing their migration into the CNS [4]. This mechanism underlies its effectiveness in reducing the frequency of MS relapses and slowing disease progression. However, the impact of fingolimod on the immune system is multifaceted and extends beyond lymphocyte sequestration. Fingolimod’s primary mechanism of action is the modulation of S1P receptors, predominantly S1P1, on lymphocytes. S1P is a bioactive lipid that plays a crucial role in regulating lymphocyte egress from lymphoid tissues. By acting as a functional antagonist at these receptors, fingolimod causes the internalization and subsequent degradation of the S1P1 receptor, effectively trapping lymphocytes in lymph nodes [5,6]. This reduction in circulating lymphocytes, particularly central memory T cells, is thought to be central to its efficacy in reducing MS relapses [7].

However, fingolimod also interacts with other S1P receptor subtypes expressed on various cells, including neural cells, suggesting its potential direct effects on the CNS. Studies have indicated that fingolimod may have neuroprotective properties, possibly through mechanisms such as reducing neuroinflammation, promoting remyelination, and direct effects on neural cells [8,9]. In addition, the impact of fingolimod on the immune system is not limited to T cells. It also affects B cells, NK cells, and dendritic cells, albeit to varying extents and through different mechanisms [10]. Fingolimod primarily affects CD4+ and CD8+ T cells, reducing their numbers in the peripheral blood. This reduction is more pronounced in CCR7+ central memory T cells compared to effector memory T cells, which are less dependent on S1P1-mediated egress [11]. This selective sequestration can modulate the autoimmune response against myelin antigens. Fingolimod also influences B cell dynamics, although the effects are less pronounced than on T cells. It can alter B cell subpopulations, potentially impacting antibody production and B cell-mediated autoimmunity [12]. Furthermore, the impact on NK cells is complex, with reports suggesting both their reduction and functional alteration. NK cells play a role in immune surveillance, and their modulation with fingolimod could have implications for infection risk and tumor surveillance [13]. Fingolimod may also affect dendritic cell migration and function, potentially influencing antigen presentation and T cell activation [14].

While fingolimod has demonstrated efficacy in reducing MS relapse rates and slowing disease progression, its immunomodulatory effects raise concerns regarding infection risk and immune surveillance. Patients on fingolimod have an increased risk of infections, including opportunistic infections such as the varicella-zoster virus (VZV) [1]. This necessitates careful monitoring and management strategies, including vaccination, before therapy initiation. Moreover, there have been reports of progressive multifocal leukoencephalopathy (PML), a rare but serious brain infection, in patients taking fingolimod [15]. This underscores the need for vigilant monitoring and a thorough understanding of the risk factors associated with fingolimod use.

Overall, fingolimod represents a significant advancement in the treatment of MS, offering a mechanism of action through S1P receptor modulation. Its impact on the immune system is complex, affecting multiple cell types and functions. While it provides substantial benefits in controlling MS disease activity, it also poses challenges in terms of infection risk and immune surveillance. Ongoing research and clinical experience continue to refine our understanding of fingolimod’s immunological effects, guiding its optimal use in MS management.

In MS research, assessing a patient’s immunological status, disease progression, and treatment efficacy is crucial [2]. Produced by activated T lymphocytes and NK cells, IFN-γ, called the master regulator of immunity, plays a significant role in immunomodulation and inflammation, acting as a critical prognostic factor in MS. Elevated IFN-γ levels are associated with worsening MS symptoms and exacerbations and are prevalent in MS lesions. It is believed to contribute to oligodendrocyte apoptosis and is found in increased quantities in mononuclear cells and the cerebrospinal fluid of MS patients [16].

The QuantiFERON^®^-Monitor (QFM, Qiagen, Hilden, Germany) is an innovative blood assay measuring IFN-γ production after stimulating both innate and adaptive immune cells, offering an advantage over single-stimulant assays. This test, requiring minimal laboratory processing, is based on the widely used QMF Gold assay kit platform [17,18]. While previously employed in fields such as transplantation, infectious medicine, and hepatology, its application in neurology, especially for multiple sclerosis patients, marks a novel approach. In our previous work, we were the first to use this monitor for evaluating immune functions in MS patients, breaking new ground in the assessment and understanding of MS [19].

The objective of this research was to undertake QFM assessments for individuals with MS, aimed at assessing the immune system’s condition by evaluating a range of specific cytokines in stimulated blood samples to gain deeper insights into the immune reactions. Building on the findings from a preceding publication [19], which employed activated serum samples initially for determining IFN-γ, our present investigation shifted its focus toward a broader array of cytokines to uncover further details about the immune response. We employed Luminex technology for an initial screening, leading to intriguing results, which were then meticulously validated through ELISA, offering a nuanced exploration of cytokine patterns. Our prior study centered on contrasting the levels of IFN-γ among MS patients, untreated progressive MS (pMS) patients, and healthy controls (HCs), uncovering a reduced immune activation in MS patients, especially pronounced in those treated with fingolimod, a known suppressor of IFN-γ production. The markedly lower QFM scores in MS patients compared to HCs point to an impaired baseline immune state in MS. In contrast to an earlier study, the current research broadens the monitoring to a comprehensive array of immune markers using QFM, followed by an extensive evaluation of selected inflammatory markers through Luminex technology (IFN-α, IFN-γ, IL-1α, IL-1β, IL-1ra, IL-2, IL-3, IL-4, IL-6, IL-7, IL-10, IL-15, IL-33, VEGF), and substantiating these promising indicators with the ELISA technique (IL-1α, IL-1ra, IL-6, VEGF), thus offering an expansive overview of the immune landscape in MS.

## 2. Results

### 2.1. Characteristics of Previous Treatment

Table 1 presents a detailed breakdown of the previous treatments received by the study participants with multiple sclerosis, including pMS and fMS. These treatments are categorized by drug type, with the number of patients and their respective percentages in the overall group, pMS subgroup, and fMS subgroup.

### 2.2. Demographics and Characteristics of Healthy Controls

Table 2 provides the key characteristics of the healthy control group (N = 86) in the study, offering insights into their demographic and biological markers. A significant majority of the healthy controls were female, comprising 87.2% (75 out of 86) of the group. The entire group (100%) had an average age of 49.31 years with a median age of 48 years, indicating a middle-aged cohort. All participants (100%) were analyzed for IL-1ra levels, with an average of 87,061.42 pg/mL and a median of 71,120 pg/mL. Nearly all participants (98.8%) had measurable IL-1α levels, averaging 137.77 pg/mL with a median of 119.8 pg/mL. VEGF levels were recorded in 95.3% of the group, with an average of 123.04 pg/mL and a median of 39.41 pg/mL. Also, 95.3% of the group had measurable IL-6 levels, averaging 36,785.23 pg/mL with a median of 35,805 pg/mL.

### 2.3. Demographics and Characteristics of Progressive MS Patients

Table 3 details the characteristics of the progressive multiple sclerosis group (N = 96) in this study, providing information on their demographics and various biological markers. In this group, females constituted 76.0% (73 out of 96) of the participants, reflecting a significant female predominance. The entire group (100%) had an average age of 57.57 years, with a median of 58.5 years. This suggests a relatively older cohort compared to the general MS population, which aligns with the progressive nature of pMS. The average EDSS (Expanded Disability Status Scale) score was 6.09, with a median of 6.5, indicating a relatively high level of disability, which is expected in progressive MS. The average duration of the disease in this group was 18.30 years, with a median of 17 years, showing the chronic aspect of this condition. All participants (100%) were analyzed for IL-1ra levels, with an average of 111,563.58 pg/mL. All participants had measurable IL-1α levels, averaging 89.39 pg/mL. The wide range in values (from 0 to 146) indicated variability in the inflammatory response among individuals. Nearly all participants (97.9%) had their VEGF levels recorded, with an average of 134.75. Also, 97.9% of the group had measurable IL-6 levels, averaging 36,248.52 pg/mL.

### 2.4. Demographics and Characteristics of Fingolimod-Treated MS Patients

Table 4 provides an overview of the characteristics of the group of multiple sclerosis patients treated with fingolimod (N = 98), offering insights into their demographics and various biological markers. In this group, 62.2% (61 out of 98) of the participants were female, showing a female majority but less pronounced than the general MS population. The entire group (100%) had an average age of 41.80 years, with a median of 43 years, indicating a younger cohort compared to the progressive MS group. The average EDSS score was 4.04, with a median of 4.50, suggesting a moderate level of disability, lower than the progressive MS group. The average duration of the disease in this group was 11.66 years, with a median of 11 years, which was shorter than the progressive MS group. The average duration of fingolimod treatment is 4.59 years, with a median of 5 years, reflecting long-term treatment in this cohort. All participants (100%) were analyzed for IL-1ra levels, with a high average of 131,064.55 pg/mL. This suggests significant anti-inflammatory activity. All participants had measurable IL-1α levels, averaging 179.94 pg/mL. This higher average compared to other groups may reflect specific immune modulation due to fingolimod treatment. All participants had their VEGF levels recorded, averaging 97.66 pg/mL, which is lower than the progressive MS group. Also, 100% of the group had measurable IL-6 levels, averaging 46,665.78 pg/mL.

### 2.5. Interleukin Levels

The presented Violin plots (Figure 1, Figure 2, Figure 3 and Figure 4) illustrate the distribution of values for four cytokines (IL-1α, VEGF, IL-1ra, and IL-6) across the following three groups: HC, pMS, and fMS.

IL-1α (Figure 1): healthy controls show a wider spread of lower values and a longer tail of higher values. Patients with pMS and fMS have higher median values, suggesting increased levels of IL-1α in these groups.

VEGF (Figure 2): The distribution is similar to IL-1α but with a significant tail of high values in pMS, which may indicate large individual differences within this group.

IL-1ra (Figure 3): All groups display a similar distribution width with high IL-1ra values, yet patients with fMS have slightly higher median values than the other groups.

IL-6 (Figure 4): The distribution of values is fairly consistent across all groups, but the fMS group has somewhat higher median values, which could indicate an increased immune activity or inflammation in these patients.

### 2.6. Pairwise Comparison of the Interleukin Levels

Table 5 presents a pairwise comparison of interleukin levels between different groups: healthy controls, pMS, and fMS. The *p*-values indicate the statistical significance of differences between these groups. Gender (the number of females %): there are statistically significant differences in gender distribution between HC and fMS (*p* < 0.0001) and between pMS and fMS (*p* = 0.04), but not between NC and cMS (*p* = 0.06). There are highly significant age differences across all comparisons: HC vs. pMS (*p* < 0.0001), HC vs. fMS (*p* < 0.0001), and pMS vs. fMS (*p* < 0.0001). The difference in disability status between pMS and fMS is extremely significant (*p* < 0.0001). The duration of the disease shows a highly significant difference between pMS and fMS (*p* < 0.0001). Significant differences in IL-1ra levels are observed between HC and fMS (*p* = 0.02), HC and fMS (*p* < 0.0001), and pMS and fMS (*p* = 0.04). There are significant differences in IL-1α levels between all groups: HC vs. pMS (*p* = 0.01), HC vs. fMS (*p* = 0.03), and pMS vs. fMS (*p* < 0.0001). No significant differences are found in VEGF levels between the groups. There are significant differences in IL-6 levels between HC and fMS (*p* = 0.00059) and pMS and fMS (*p* = 0.000593), but not between HC and pMS (*p* = 0.84).

### 2.7. Logistic Regression with Prediction of Belonging to Groups “HC = 0” and “pMS = 1”

Table 6 presents the results of a logistic regression analysis aimed at predicting the likelihood of belonging to either the healthy control group (HC = 0) or the progressive multiple sclerosis group (pMS = 1). The analysis used 182 observations with seven predictors: a constant, levels of IL-1ra, IL-1α, VEGF, IL-6, age, and gender.

Table 7 displays the results from a logistic regression analysis predicting membership in either the progressive multiple sclerosis (pMS = 0) or fingolimod-treated MS (fMS = 1) groups, utilizing 194 observations.

## 3. Discussion

In this discussion, we focus on interpreting and understanding the implications of our research findings in the field of MS. Our study has provided new insights into the demographic and clinical characteristics of MS patients, including differences in cytokine levels and treatment regimens. Special attention is given to analyzing the relationship between various treatment approaches and the clinical characteristics of these patients, as well as the impact of these factors on the progression and management of MS. The discussion also reflects on the significance of these findings for future research and clinical practice.

The data summarized in Table 1 provide a comprehensive demographic and clinical profile of the participants involved in a multiple sclerosis study. The predominance of women in this study (74.6%) reflects the known higher incidence of MS in females. The mean age of participants (49.51 years) and the average disease duration (15.43 years) highlight the chronic nature of MS and suggest that the sample includes individuals who are likely to be at various stages of disease progression. The average EDSS score of 5.21 across 60.4% of the patients indicates a moderate to severe level of disability, which underscores the significant impact MS has on daily functioning and quality of life for a substantial portion of this study’s population. Notably, 26.1% of the patients underwent treatment with fingolimod, an immunomodulatory therapy, for an average duration of 4.59 years. This provides an opportunity to analyze the long-term effects of this treatment on disease progression and immune response.

Table 2 offers valuable insight into the treatment landscape of MS prior to the study. The use of dimethyl fumarate was notably low, suggesting it may not be the first-line treatment for the majority of these patients, or it could reflect selection bias within the study population. Glatiramer acetate (GA), on the other hand, was more commonly administered, indicating its established role in MS treatment protocols. The combination therapies involving GA and interferon, as well as GA and teriflunomide, albeit utilized by a small fraction of patients, highlight the complexity of managing MS, where monotherapy may not suffice, and combination therapies are employed to target different aspects of the disease mechanism. IFN stands out as the most frequently used treatment, which may be attributed to its long-standing presence in the field of MS treatment and its efficacy profile. This table underscores a tailored approach to MS treatment, reflecting individual patient needs, disease severity, and progression, as well as responses to previous treatments. It also signals the dynamic nature of MS therapy, where newer treatments like fingolimod are incorporated into the therapeutic arsenal, offering hope for improved disease management.

Cytokine profile data (Table 3, Table 4 and Table 5 and Figure 1, Figure 2 and Figure 3), including IL-1ra, IL-1α, VEGF, and IL-6, points to the complex interplay of inflammatory processes and immune responses in MS. Elevated levels of these cytokines may correlate with disease activity and could potentially serve as biomarkers for disease progression and treatment efficacy. This information is essential for understanding the pathophysiology of MS and could have implications for the development of targeted therapies. This study’s findings warrant further investigation into the role of these cytokines in MS and their utility in clinical practice.

In our research, logistic regression was employed to identify and quantify the relationships between various variables associated with multiple sclerosis. Specifically, regression analysis enabled us to perform the following: (a) Through logistic regression, we could ascertain which factors (such as cytokine levels, age, gender, and duration of the disease) significantly correlate with whether a patient belongs to the progressive MS group or the fingolimod-treated group; (b) the coefficients calculated within the regression model provided insights into the strength of the relationships between independent variables (e.g., cytokines) and the dependent variable (group membership). This allowed us to understand which factors exert a greater influence on the differences between patient groups; and (c) predict the risk or probability of a certain condition. The convergence of the model indicates that the maximum likelihood estimation procedure was successfully completed, finding a viable solution. The small standard errors associated with each coefficient instill confidence in the reliability of the estimates. Consequently, this logistic regression model proves to be an effective instrument for discerning the variables that distinguish HC and pMS, with certain cytokine levels and demographic factors contributing significantly. The regression analysis model demonstrated robust convergence, as evidenced by the minimal standard errors of the coefficients, which substantiates the dependability of the estimates. These outcomes imply that specific cytokines, in conjunction with age and EDSS scores, play a pivotal role in distinguishing between patients with progressive MS and those undergoing treatment with fingolimod.

In the case of the prediction of belonging to groups “HC = 0” and “pMS = 1”, the model’s Pseudo R-squared value of 0.2133 indicates a reasonable fit to the data, explaining approximately 21.33% of the variance in the group membership. The log-likelihood value at −99.022 suggests that the model’s estimation of the predicted probabilities fits the observed outcomes quite well compared to the null model, which has a log-likelihood of −125.88. The coefficients for IL-1ra and age are statistically significant (*p* < 0.05), indicating they are strong predictors in distinguishing between HC and pMS. Specifically, the IL-1ra has a positive coefficient, suggesting that higher levels increase the likelihood of being in the pMS group, while age also shows a positive relationship, meaning that older participants are more likely to be classified as pMS. Conversely, IL-1α and gender have negative coefficients that are statistically significant, indicating that higher levels of IL-1α and being female reduce the likelihood of being in the pMS group. VEGF and IL-6 did not show statistical significance in this model (*p* > 0.05), suggesting that these variables do not have a strong differential effect on the likelihood of belonging to one group over the other within this dataset.

Also, in the case of the prediction of belonging to groups “pMS = 0” and “fMS = 1”, the model demonstrated a strong fit, as indicated by a Pseudo R-squared value of 0.6560, suggesting that it explains a significant portion of the variance between the two groups. The log-likelihood value of −46.260 was considerably improved from the null model’s log-likelihood of −134.46, further evidencing this model’s effectiveness. Statistically significant predictors include the following: IL-1ra, which has a positive coefficient, indicating that higher levels are associated with a greater likelihood of being in the fMS group (*p* = 0.029). IL-1α also has a positive coefficient and is significant (*p* = 0.003), suggesting its levels are higher in the fMS group. Age shows a negative coefficient (*p* = 0.000), meaning older participants are less likely to be in the fMS group. EDSS has a substantially negative coefficient (*p* = 0.000), indicating that higher disability scores are associated with lower odds of being in the fMS group. Other variables like VEGF, IL-6, gender, and MS duration did not reach statistical significance, suggesting they do not have a discernible impact on distinguishing between the pMS and fMS groups in this dataset.

The limitations of these data may include several aspects. Selection bias, where participants were not randomly selected, could lead to results that are not representative of the entire population of patients with multiple sclerosis (MS). Additionally, there is a risk of gender imbalance, as the predominance of women in the study might influence the interpretation of the results, especially since MS can manifest differently across genders. A significant limitation is also the focus of this study on only a few cytokines, which do not encompass the entire spectrum of immune markers. These limitations must be considered when interpreting the results and planning future research.

Despite the limitations of this study, our findings provide valuable insights into the demographic and clinical aspects of multiple sclerosis, highlighting the importance of cytokines. Our observations, particularly regarding cytokines like IL-1ra, IL-1α, VEGF, and IL-6, enhance our understanding of MS pathophysiology and pave the way for advancements in MS research and clinical management.

## 4. Materials and Methods

### 4.1. Study Population

This observational study was conducted from August 2019 to June 2023. In total, 280 participants were enrolled in the study. The study group comprised 86 HC, 96 pMS, and 98 relapse-remitting multiple sclerosis (RRMS) fMS patients.

The inclusion criteria for healthy controls were in the age range of 18 and 75 years.

The inclusion criteria for patients with pMS were as follows:The participant had to be between 18 and 75 years of age inclusive of the time of signing the informed consent.The participant had to be previously diagnosed with RRMS in accordance with the 2017 revised McDonald criteria [20].The participant had to be currently diagnosed with SPMS in accordance with the clinical course criteria1, revised in 2013 [21].Absence of clinical relapses for at least 24 months.No administration of any of the following drugs: intravenous immunoglobulin, dimethyl fumarate, fingolimod, teriflunomide, azathioprine, mycophenolate mofetil, methotrexate, and B-cell depleting therapies, such as ocrelizumab and rituximab 12 months prior the entry to the study; mitoxantrone, cyclophosphamide, cladribine, cyclosporine, and alemtuzumab 2 years prior the entry to the study; and treatment with methylprednisolone, glatiramer acetate, and interferon beta 3 months prior the entry to the study.

The inclusion criteria for patients with RRMS treated with fingolimod were as follows:The participant had to be between 18 and 75 years of age, inclusive at the time of signing the informed consent.The participant had to be diagnosed with RRMS in accordance with the 2017 revised McDonald criteria [20].Absence of clinical relapses for at least 12 months.Treatment with fingolimod for at least 3 years.

All patients signed written informed consent. The study protocol was approved by the Ethics Committee of the University Hospital Hradec Kralove (Reference Number 202009P05; 10 September 2020).

### 4.2. Screening Multiplex Examination of Cytokines

Initially, a comprehensive assessment of immune markers was conducted using Luminex technology (IFN-α, IFN-γ, IL-1α, IL-1β, IL-1ra, IL-2, IL-3, IL-4, IL-6, IL-7, IL-10, IL-15, IL-33, VEGF), which allowed for a broad evaluation of multiple inflammatory markers simultaneously. Following this extensive screening, promising values were then meticulously analyzed using the more detailed ELISA method (IL-1α, IL-1ra, IL-6, VEGF), providing a focused and in-depth understanding of specific cytokines of interest. The findings and implications of these cytokine levels from the ELISA tests are subsequently discussed in this article, offering an extensive evaluation of their significance in the context of multiple sclerosis.

The human cytokine luminex^®^ performance assay was chosen for the screening analysis of selected cytokines: IFN-α, IFN-γ, IL-1α, IL-1β, IL-1ra, IL-2, IL-3, IL-4, IL-6, IL-7, IL-10, IL-15, IL-33 and VEGF (R&D systems, Minneapolis, MN, USA). Samples of activated sera came from a previous study aimed at determining levels of IFN-γ using the functional QuantiFERON-Monitor test (Qiagen, Hilden, Germany) in patients with MS. The QuantiFERON-Monitor assay uses a combination of lyophilized stimulants containing R848 and CD3 antibodies (QuantiFERON Monitor LyoSpheres™, Qiagen, Hilden, Germany), which are added to 1 mL of heparinized whole blood within 8 h after collection. Such stimulation ensures the non-specific stimulation of cells of innate and adaptive immunity and the enhancement of their cytokine production. Stimulated blood samples were incubated for 16 to 24 h at 37 °C and then, after centrifugation, samples of serum were collected, aliquoted, and stored at −80 °C until further analysis. Cytokine levels were determined using Luminex xMAP^®^ magnetic technology by Bio-Plex 200 system (Bio-Rad, Hercules, CA, USA), and an analysis of the results was performed using xPONENT^®^ 4.2. analysis software (Luminex Corporation, Austin, TX, USA). The assay was run according to the instructions for use provided by the manufacturer. All values were expressed in picograms (pg) per milliliter of the serum.

### 4.3. IL-1α, IL-1ra, IL-6, and VEGF Detection using ELISA

The promising results obtained from the screening examination were subsequently verified by a more sensitive enzyme-linked immunosorbent assay technique (ELISA) using ELISA kits for human IL-1α, IL-1ra, IL-6, and VEGF (R&D systems, Minneapolis, MN, USA) according to the manufacturer’s instructions. The minimal detectable doses of selected kits were 0.1 pg/mL for IL-1α, 2.2 pg/mL for IL-1ra, 0.7 pg/mL for IL-6, and 5.0 pg/mL for VEGF. Absorbance was read at 450 nm using a Multiskan RC ELISA reader (Thermo Fisher Scientific, Waltham, MA, USA). All values were expressed in pg per milliliter of the serum.

### 4.4. Statistical Analysis

The study included a total sample of 280 individuals, categorized into three groups: normal controls (N = 86), chronic MS (N = 96), and MS treated with fingolimod (N = 98). The levels of interleukins IL-1ra, IL-1α, VEGF, and IL-6 were summarized using the mean, standard deviation, median, lower quartile, and upper quartile. Categorical variables, such as gender and previous treatment, were described by frequency and percentage.

Parametric testing was employed for post hoc pairwise comparisons. Specifically, *t*-tests with a Tukey correction were utilized to adjust for the false discovery rate. This choice was predicated on the data’s adherence to the assumptions required for parametric tests, namely normal distribution, sample independence, and the homogeneity of variances. Logistic regression analysis was conducted using a Generalized Linear Model (GLM) framework, employing a binomial distribution and a logit link function. This approach is particularly apt for the binary nature of our outcome variable, which categorizes group membership into healthy controls (HC) or patients with progressive multiple sclerosis (pMS).

Furthermore, to address potential deviations from normality, the Iteratively Reweighted Least Squares (IRLS) method was applied due to its robustness in parameter estimation. Our analytical strategy was carefully selected to provide the most precise and reliable interpretation of data, which encompassed a variety of covariates, including age, gender, EDSS score, and disease duration. Through this comprehensive and methodologically sound approach, we aimed to ensure the integrity and accuracy of our findings.

### 4.5. Normality Assessment

The normality of the interleukin levels was assessed graphically and tested formally using the Liliefors test for normality.

### 4.6. Comparison of Interleukin Levels

The levels of interleukins IL-1ra, IL-1α, VEGF, and IL-6 were compared between the groups using the following methods: Gender: the χ^2^ test for independence in contingency tables. Age: a parametric ANOVA. Expanded Disability Status Scale between MS patient groups: a parametric *t*-test. Post hoc pairwise comparison: parametric testing using a *t*-test with Tukey correction for the false discovery rate.

### 4.7. Logistic Regression Analysis

A logistic regression analysis was conducted to predict group membership based on the levels of interleukins IL-1ra, IL-1α, VEGF, and IL-6. The analysis utilized a Generalized Linear Model (GLM) with a binomial family and logit link function. The method employed was Iteratively Reweighted Least Squares (IRLS), and the model was non-robust. The regression model included covariates such as age, gender, EDSS, and disease duration.

### 4.8. Multiple Testing and Significance Level

All *p*-values were corrected for multiple testing using the Benjamini–Hochberg procedure. Statistical significance was defined at the 5% level using a two-tailed alternative hypothesis.

### 4.9. Software

All statistical computations were performed using Python libraries Numpy, Statmodels, and Pandas (Python version 3.10.9) [22].

## 5. Conclusions

Our comprehensive study on MS has yielded significant insights into the demographic and clinical characteristics of MS patients, highlighting the role of cytokines and treatment regimens in disease progression. The predominance of women and the varied stages of disease progression within our study group underscores the complexity of MS. We observed that treatments like fingolimod and interferon are key components in managing MS, reflecting the need for personalized treatment approaches. Our findings on cytokine levels, particularly IL-1ra, IL-1α, VEGF, and IL-6, suggest their potential as biomarkers and their role in the pathophysiology of MS. The results show that patients with fMS after stimulation show the highest levels of all tested analytes of the compared groups, except for VEGF, where the level was, on the contrary, the lowest. However, while VEGF levels remained without statistically significant changes, in the case of IL-6, a higher concentration was evident in this group of patients, both compared to HC and patients with pMS. The group of patients with fMS also showed statistically significantly higher levels of the cytokine IL-1α and its receptor antagonist than the compared groups. However, this difference was also noticeable between HC and pMS. While IL-1 receptor antagonist production is lower in HC than pMS, the production of IL-1α alone is statistically higher. It is evident from the results that the cytokine profile of the patients, or rather the ability of cytokine production after the non-specific stimulation of their immunity, is significantly influenced by the selected treatment approach and the state of the disease. From an immunological point of view, it appears to be good news that the significantly manipulated immune system of patients with fMS is still capable of a significant response to stimulation, and it can, thus, be assumed that it does not remain defenseless even in case of possible infection. Likewise, the fact that the reactivity of pMS patients who have not received significantly immune-compromising treatment is not objectively dramatically worse than the reactivity of HC appears to be a success.

## Figures and Tables

**Figure 1 ijms-25-02179-f001:**
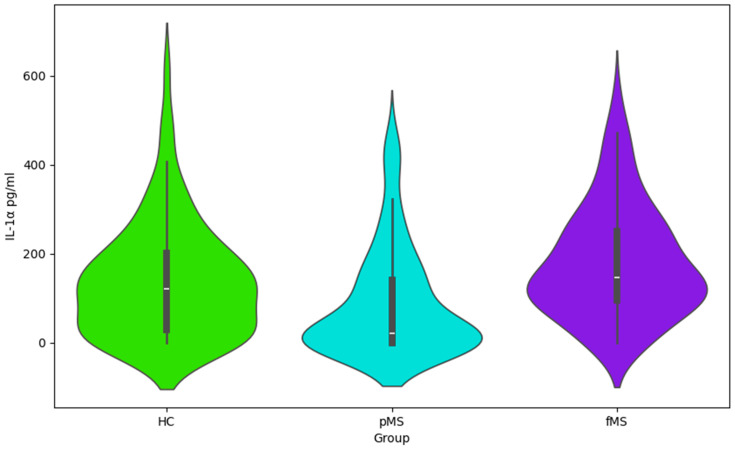
Violin plot for IL-1a values by group.

**Figure 2 ijms-25-02179-f002:**
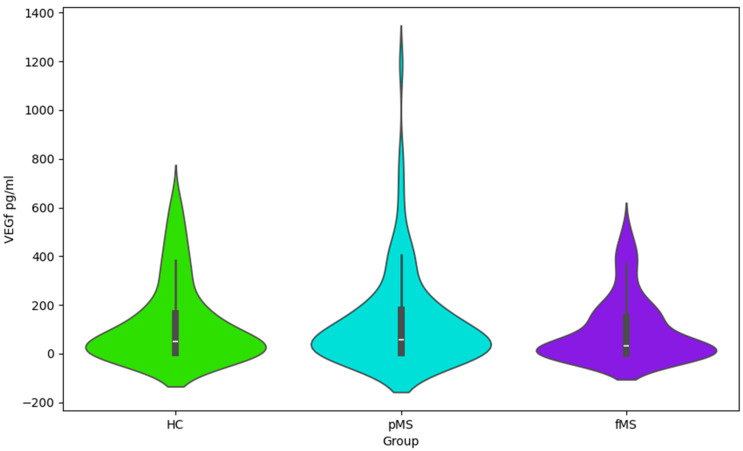
Violin plot for VEGF values by group.

**Figure 3 ijms-25-02179-f003:**
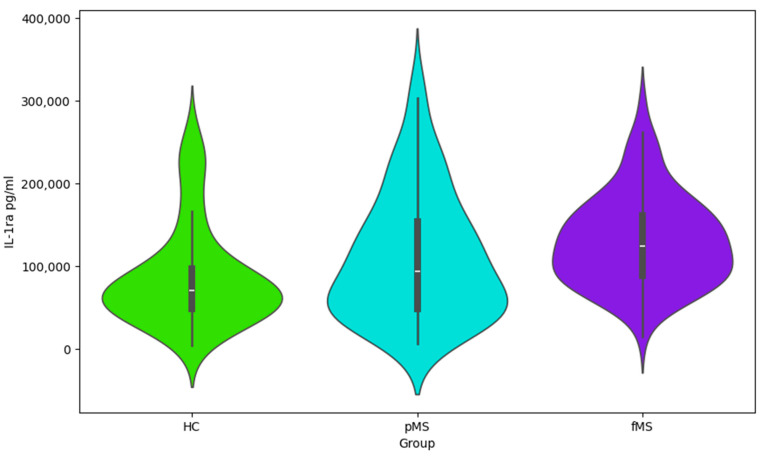
Violin plot for IL-1ra values by group.

**Figure 4 ijms-25-02179-f004:**
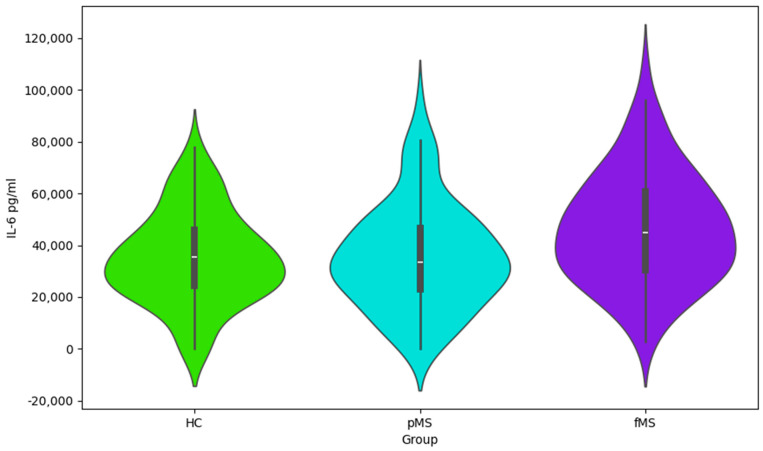
Violin plot for IL-6 values by group.

**Table 1 ijms-25-02179-t001:** Characteristics of previous treatment.

	Overall	Progressive MS	MS Treated with Fingolimod
DMF	1 (0.4)	0	1 (0.4)
GA	20 (7.1)	0	20 (7.1)
GA, IFN	2 (0.7)	0	2 (0.7)
GA, teriflunomid	1 (0.4)	0	1 (0.4)
IFN	36 (12.9)	1 (0.4)	35 (12.5)
IFN, DMF	2 (0.7)	0	2 (0.7)
IFN, GA	3 (1.1)	0	3 (1.1)
Teriflunomid	1 (0.4)	0	1 (0.4)

Note: DMF—Dimethyl fumarate; GA—Glatiramer acetate; INF—interferon.

**Table 2 ijms-25-02179-t002:** Demographics and characteristics of healthy controls (N = 86).

Characteristics	N (%)	Mean (SD)	Median (p25, p75)
Gender—number of females (%)	75 (87.2)		
Age	86 (100)	49.31 (9.65)	48 (44, 55.25)
IL-1ra (pg/mL)	86 (100)	87,061.42 (60,289.28)	71,120 (48,685, 99,315)
IL-1α (pg/mL)	85 (98.8)	137.77 (126.93)	119.8 (24.35, 204.4)
VEGF (pg/mL)	82 (95.3)	123.04 (166.82)	39.41 (0.41, 175.55)
IL-6 (pg/mL)	82 (95.3)	36,785.23 (16,152.26)	35,805 (24,330, 46,850)

Note: VEGF—vascular endothelial growth factor, SD—standard deviation, p25—25th percentile, p75—75th percentile.

**Table 3 ijms-25-02179-t003:** Demographics and characteristics of progressive MS patients (N = 96).

Characteristics	N (%)	Mean (SD)	Median (p25, p75)
Gender—number of females (%)	73 (76.0)		
Age	96 (100)	57.57 (8.90)	58.5 (51.25, 64.75)
EDSS	96 (100)	6.09 (1.01)	6.5 (5.5, 6.5)
Disease duration	96 (100)	18.30 (6.65)	17.00 (12.00, 23.00)
Fingolimod treatment duration (in years)	NA	NA	NA
IL-1ra (pg/mL)	96 (100)	111,563.58 (75,575.09)	93,440 (48,632.50, 156,650)
IL-1α (pg/mL)	96 (100)	89.39 (120.11)	21.17 (0, 146)
VEGF (pg/mL)	94 (97.9)	134.75 (197.01)	58.98 (3.75, 184.63)
IL-6 (pg/mL)	94 (97.9)	36,248.52 (19,623.21)	33,915 (23,227.50, 47,312.50)

Note: EDSS—Expanded Disability Status Scale, VEGF—vascular endothelial growth factor, SD—standard deviation, p25—25th percentile, p75—75th percentile.

**Table 4 ijms-25-02179-t004:** Demographics and characteristics of fingolimod-treated MS patients (N = 98).

Characteristics	N (%)	Mean (SD)	Median (p25, p75)
Gender—number of females (%)	61 (62.2)		
Age	98 (100)	41.80 (9.00)	43 (37.75, 47)
EDSS	98 (100)	4.04 (1.70)	4.50 (2.50, 5.50)
Disease duration	98 (100)	11.66 (6.04)	11.00 (6.00, 15.00)
Fingolimod treatment duration (in years)	98 (100)	4.59 (1.96)	5.00 (3.00, 6.00)
IL-1ra (pg/mL)	98 (100)	131,064.55 (54,638.82)	124,750.00 (88,637.50, 162,600.00)
IL-1α (pg/mL)	98 (100)	179.94 (124.75)	138.95 (93.80, 252.90)
VEGF (pg/mL)	98 (100)	97.66 (131.73)	27.06 (0, 152.63)
IL-6 (pg/mL)	98 (100)	46,665.78 (21467.17)	45,065, (30,092.50, 62,062.75)

Note: EDSS—Expanded Disability Status Scale, VEGF—vascular endothelial growth factor, SD—standard deviation, p25—25th percentile, p75—75th percentile.

**Table 5 ijms-25-02179-t005:** Pairwise comparison of the interleukin levels.

Characteristics	*p*-Value (HC–pMS)	*p*-Value (HC–fMS)	*p*-Value (pMS–fMS)
Gender—number of females (%)	0.06	<0.01	0.04
Age	<0.01	<0.01	<0.01
EDSS			<0.01
Disease duration			<0.01
IL-1ra	0.02	<0.01	0.04
IL-1α	0.01	0.03	<0.01
VEGF	0.67	0.27	0.13
IL-6	0.84	<0.01	<0.01

Note: The levels of interleukins IL-1ra, IL-1α, VEGF, IL-6, age, EDSS, and disease duration were compared between the groups using the independent samples *t*-test. Differences between genders were evaluated using the χ^2^ test for independence in contingency tables. Post hoc pairwise comparison: parametric testing using a *t*-test with Tukey correction for the false discovery rate. EDSS—Expanded Disability Status Scale, VEGF—vascular endothelial growth factor, HC—healthy control, fMS—fingolimod-treated MS, pMS—progressive multiple sclerosis.

**Table 6 ijms-25-02179-t006:** Logistic regression analysis predicting group membership (healthy control, HC = 0; progressive multiple sclerosis, pMS = 1).

Variable	Coefficient	Std. Error	z-Value	*p* > |z|	95% CI
Constant	−4.543	1.133	−4.011	<0.001	[−6.764, −2.324]
IL-1ra	5.63 × 10^−6^	2.72 × 10^−6^	2.071	0.038	[3.01 × 10^−7^, 1.1 × 10^−5^]
IL-1α	−0.005	0.002	−3.041	0.002	[−0.008, −0.002]
VEGF	−2 × 10^−4^	0.001	−0.169	0.866	[−0.002, 0.002]]
IL-6	−1.33 × 10^−5^	1.07 × 10^−5^	−1.240	0.215	[−3.44 × 10^−5^, 7.73 × 10^−6^]
Age	0.110	0.021	5.208	<0.001	[0.068, 0.151]
Gender	−0.938	0.470	−1.998	0.046	[−1.858, −0.018]

Note: The *p*-values indicate the level of statistical significance for each coefficient, with values <0.001 denoted as highly significant. The 95% CI column provides the range within which the true coefficient value is expected to lie with 95% confidence.

**Table 7 ijms-25-02179-t007:** Logistic regression analysis results for group membership prediction (progressive multiple sclerosis, pMS = 0; fingolimod-treated MS, fMS = 1).

Variable	Coefficient	Std. Error	z-Value	*p* > |z|	95% CI
Constant	16.934	2.924	5.792	<0.001	[11.204, 22.664]
IL-1ra	1.18 × 10^−5^	5.41 × 10^−6^	2.179	0.029	[1.19 × 10^−6^, 2.24 × 10^−5^]
IL-1α	0.008	0.003	2.980	0.003	[0.003, 0.013]
VEGF	−5.699 × 10^−5^	0.002	−0.029	0.977	[−0.004, 0.004]
IL-6	1.87 × 10^−5^	1.65 × 10^−5^	1.138	0.255	[−1.36 × 10^−5^, 5.11 × 10^−5^]
Age	−0.247	0.043	−5.790	<0.001	[−0.330, −0.163]
Gender	−0.240	0.622	−0.386	0.700	[−1.459, 0.979]

Note: The *p*-values indicate the level of statistical significance for each coefficient, with values <0.001 denoted as highly significant. The 95% CI column provides the range within which the true coefficient value is expected to lie with 95% confidence.

## Data Availability

The data are available upon the request from the corresponding author.

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
