# Peer review of "Quantiferon Monitor Testing Sheds Light on Immune System Disparities between Multiple Sclerosis Patients and Healthy Individuals"

_ijms, 2024, doi:10.3390/ijms25042179_

Round 1

Reviewer 1 Report

Comments and Suggestions for Authors

Ilona Součková et al., in " Quatiferon Monitor Testing Sheds Light on Immune System

Disparities Between MS Patients and Healthy Individuals " provide a comparative study concerning QFM assessments for individuals with MS, aimed at assessing the immune system's condition by inducing a non-specific reaction in leukocytes and then measuring the quantity of IFN-γ emitted from these activated cells.

They outline that the have provided valuable insights concerning the importance of cytokines IL-1ra, IL-1α, VEGF, and IL-6 in MS research and clinical management.

Despite the potential interesting topic, the paper suffers from some issues related to the methodology used in data analysis and discussion of the results.

Here are some comments to consider in order to improve the content of this manuscript:

11.  In the Results section, there is a paragraph („Initially, a comprehensive assessment of immune markers,.....”, lines 133-141)  inappropriate  for this section (it has no reference to any result); it is a more appropriate paragraph for the Material and Method section

22.  Authors stated that “Table 1. summarizes the data of participants in a study on multiple sclerosis (MS)” so all the statistics presented in this table represent subjects’ characteristics  from two different/heterogeneous populations (HC and MS patients); we consider it appropriate to present the statistics for each population (as shown in the following tables), so Table 1 is unnecessary

33. Table 3, Table 4 and Table 5 need a legend (please explain: SD, p25, p75)

44. In the Figures 1-4 , the Oy axis should contain the units of measure of studied variables

55. Figures 1-4, should have a corresponding title to be positioned after the caption of figures

66. P-values form Table 6 represent probabilities so they should be presented as numeric (not scientific) format

77.  the authors did not mention either in the Statistical analysis or in the legend of Table 6, by which statistical method (which test) they performed pairwise comparisons

88. Figures 5 and 6 represent screenshots from the statistical analysis program, so an inappropriate presentation for a scientific medical research

99. In the Discussion section, authors stated that „In our research, regression analysis was employed to identify and quantify the relationships between various variables associated with multiple sclerosis”. ..please see lines  297-305); these statements do not represent a precise interpretation of the results of this study (but only the general aspects of the use of logistic regression)

110.  in the case of logistic regression, the effect size for cytokines could be estimated  by calculating the adjusted OR and 95% CI.

Author Response

Dear Madam, Sir,

Comments and Suggestions for Authors

Ilona Součková et al., in " Quatiferon Monitor Testing Sheds Light on Immune Systém Disparities Between MS Patients and Healthy Individuals " provide a comparative study concerning QFM assessments for individuals with MS, aimed at assessing the immune system's condition by inducing a non-specific reaction in leukocytes and then measuring the quantity of IFN-γ emitted from these activated cells. They outline that the have provided valuable insights concerning the importance of cytokines IL-1ra, IL-1α, VEGF, and IL-6 in MS research and clinical management. Despite the potential interesting topic, the paper suffers from some issues related to the methodology used in data analysis and discussion of the results. Here are some comments to consider in order to improve the content of this manuscript:

  1. In the Results section, there is a paragraph („Initially, a comprehensive assessment of immune markers,.....”, lines 133-141)  inappropriate  for this section (it has no reference to any result); it is a more appropriate paragraph for the Material and Method section

Thank you for your feedback. The paragraph in question (lines 133-141 in the Results section) has been appropriately moved to the Methods section and yellow marked.

  1. Authors stated that “Table 1. summarizes the data of participants in a study on multiple sclerosis (MS)” so all the statistics presented in this table represent subjects’ characteristics  from two different/heterogeneous populations (HC and MS patients); we consider it appropriate to present the statistics for each population (as shown in the following tables), so Table 1 is unnecessary

Thank you for your observation. We agree that Table 1, which summarizes the data of participants in the multiple sclerosis study, is indeed redundant. The specific details about the individual studied subpopulations are already comprehensively provided in the subsequent tables. Therefore, Table 1 has been removed for clarity and conciseness.

  1. Table 3, Table 4 and Table 5 need a legend (please explain: SD, p25, p75)

Thank you for pointing this out. As per your instructions, Table 3, Table 4, and Table 5 will be corrected to include a legend explaining "SD" (Standard Deviation), "p25" (25th percentile), and "p75" (75th percentile), yellow marked. Tables have also been renumbered.

  1. In the Figures 1-4 , the Oy axis should contain the units of measure of studied variables

Thank you for your input. I can confirm that in Figures 1-4, the Y-axis now includes the units of measure for the studied variables, as per the required format.

  1. Figures 1-4, should have a corresponding title to be positioned after the caption of figures

Thank you for bringing this to our attention. The necessary adjustments have been made and the changes are highlighted in yellow for easy identification.

  1. P-values form Table 6 represent probabilities so they should be presented as numeric (not scientific) format

Thank you for your guidance. In response to the issue of maintaining readability with very small p-values, such as those extending to 26 decimal places, the table has been replaced as per your instructions. Statistically significant differences at the 1% significance level are now indicated with the symbol "< 0.01". This updated table is highlighted in yellow for clear identification.

  1. the authors did not mention either in the Statistical analysis or in the legend of Table 6, by which statistical method (which test) they performed pairwise comparisons

Thank you for your observation. Following your suggestion, a new explanation has been added to the manuscript in the legend of table 5, clarifying the statistical method used for pairwise comparisons in Table 5. This addition is highlighted in yellow for easy reference. „The levels of interleukins IL-1ra, IL-1α, VEGF, IL-6, age, EDSS and disease duration were compared between the groups using Independent Samples t-test. Differences of gender were evaluated by χ² test for independence in contingency tables. Post hoc pairwise comparison: Parametric testing using t-test with Tukey correction for the false discovery rate.”

  1. Figures 5 and 6 represent screenshots from the statistical analysis program, so an inappropriate presentation for a scientific medical research

Thank you for pointing this out. In accordance with the standards for scientific publications, Figures 5 and 6, which were screenshots from the statistical analysis program, have been replaced in the manuscript with appropriately formatted tables (Table 6. and 7.). This change ensures a more professional and suitable presentation for medical research.

  1. In the Discussion section, authors stated that „In our research, regression analysis was employed to identify and quantify the relationships between various variables associated with multiple sclerosis”. ..please see lines  297-305); these statements do not represent a precise interpretation of the results of this study (but only the general aspects of the use of logistic regression)

Thank you for your attention to detail. The statement in the Discussion section has been corrected as follows: "In our research, logistic regression was employed to identify and quantify the relationships between various variables associated with multiple sclerosis." This revision has been highlighted in yellow in the manuscript for clear visibility.

  1. in the case of logistic regression, the effect size for cytokines could be estimated  by calculating the adjusted OR and 95% CI.

Thank you for your insightful comment. I concur that the most accurate way to quantify cytokine effects in logistic regression is through adjusted odds ratios (ORs) and their respective 95% confidence intervals (CIs). To clarify, our analysis using the Python Statsmodels library did indeed produce adjusted ORs and CIs. Statsmodels automatically calculates exponentiated coefficients (exp(coef)), representing the ORs for logistic regression, along with their adjusted CIs. This adjustment considers the other variables in the model, ensuring that the ORs accurately reflect the impact of each individual cytokine on the likelihood of multiple sclerosis, independent of other factors. The results in our paper represent these adjusted values, accurately depicting the effect sizes and their statistical significance.

Reviewer 2 Report

Comments and Suggestions for Authors

The work is valuable and raises an important issue of the immune status of patients undergoing Multiple Sclerosis treatment.

The research question posed by the authors of the study is not clear and unambiguous. Is it about assessing the use of OFM or determining the immune status of patients after various types of therapy? Please specify.

The introduction is sufficient but concerns comprehensive research on MS patients and is not detailed for the current work.

The topic is original and important, but at the beginning it is necessary to clearly define what we do not know in this field, e.g. regarding the state of immunity after various types of therapies.

Please specify whether assessment after specific types of multiple sclerosis therapy has been defined and whether it is a new achievement in this field of knowledge.

The aim of the work is written quite generally and it seems that the obtained results and the discussion do not fully correspond to the assumed goal.

Please clearly define the research problem and answer this question clearly. Does the work provide new data, e.g. in relation to the described introduction and introduction to the topic. Please answer a clearly stated research question.

The work should contain a separate, detailed research question and the answer to this question along with conclusions.

The lack of a separate control group reduces the reliability of the results, but it is a clinical trial in which therapies in different groups can be compared between. In my opinion it is OK.

The description of the results is included in a restricted form. The description of the results should be developed in accordance with the main research question - the goal - and additionally with side questions.

Conclusion is general, please specify to your specific question.

Not all table descriptions in the text include the results contained in the tables. The tables with the results are not described in detail.

There is no indication of statistical significance that the authors wanted.

There is no description of why the statistical tests presented were chosen to analyze the results.

References are OK.

Author Response

Dear Madam, Sir,

Comments and Suggestions for Authors: The work is valuable and raises an important issue of the immune status of patients undergoing Multiple Sclerosis treatment. The research question posed by the authors of the study is not clear and unambiguous. Is it about assessing the use of OFM or determining the immune status of patients after various types of therapy? Please specify.

  1. The introduction is sufficient but concerns comprehensive research on MS patients and is not detailed for the current work.
  2. The topic is original and important, but at the beginning it is necessary to clearly define what we do not know in this field, e.g. regarding the state of immunity after various types of therapies.
  3. Please specify whether assessment after specific types of multiple sclerosis therapy has been defined and whether it is a new achievement in this field of knowledge.
  4. The aim of the work is written quite generally and it seems that the obtained results and the discussion do not fully correspond to the assumed goal.
  5. Please clearly define the research problem and answer this question clearly. Does the work provide new data, e.g. in relation to the described introduction and introduction to the topic. Please answer a clearly stated research question.
  6. The work should contain a separate, detailed research question and the answer to this question along with conclusions.

Thank you for your insightful comments 1-6. In response, both the introduction and conclusion have been thoroughly revised to address your concerns. These revisions include a clearer definition of the knowledge gaps in the field, particularly regarding the state of immunity after various MS therapies, a more detailed focus on the specifics of our current work, clarification of whether the assessment after specific MS therapies represents a new contribution to the field, ensuring that the aims, results, and discussion are closely aligned, and finally, articulating a clear research problem and question, with a detailed answer and conclusions. These changes are now highlighted in yellow in the manuscript for ease of review.

  1. The lack of a separate control group reduces the reliability of the results, but it is a clinical trial in which therapies in different groups can be compared between. In my opinion it is OK.

Thank you for your comment and understanding regarding the structure of our study. We appreciate your acknowledgment of the comparative approach between different therapy groups, despite the absence of a separate control group. Your feedback is highly valued.

  1. The description of the results is included in a restricted form. The description of the results should be developed in accordance with the main research question - the goal - and additionally with side questions.

Thank you for your input. Based on the requirements of both reviewers, the Results section has been revised to provide a more comprehensive description.

  1. Conclusion is general, please specify to your specific question.

Thank you for your feedback. In response, the conclusion has been revised to more specifically address our research question, ensuring it aligns closely with the focus of our study.

  1. Not all table descriptions in the text include the results contained in the tables. The tables with the results are not described in detail.

Thank you for your feedback. In response, the conclusion has been revised to more specifically address our research question, ensuring it aligns closely with the focus of our study.

  1. There is no indication of statistical significance that the authors wanted.

Thank you for your input. We appreciate attention to the presentation of statistical significance in our manuscript. We have included statements of statistical significance throughout our results section, where specific P-values are reported to demonstrate the strength of the associations observed. For instance, we have indicated the statistical significance of interleukins IL-1ra and age in the logistic regression analysis, with P-values less than 0.05, signifying their role as strong predictors in distinguishing between healthy controls (HC) and patients with progressive MS (pMS)​​. To further clarify the significance of our findings, we have defined statistical significance at the 5% level using a two-tailed alternative hypothesis and employed the Benjamini–Hochberg procedure to correct for multiple testing​​. These details are provided to assure the reader of the rigor in our analytical approach. In light of your comment, we will ensure that our final manuscript includes a clearer indication of statistical significance where necessary.

  1. There is no description of why the statistical tests presented were chosen to analyze the results.

Thank you for your comment. the Statistical analysis section has been updated to provide a more comprehensive description. Let me briefly explain, that we selected parametric testing, specifically t-tests with Tukey correction for the false discovery rate, to conduct post hoc pairwise comparisons due to the nature of our data, which meets the assumptions of parametric tests (i.e., normally distributed, independent samples, homogeneity of variances)​​. The logistic regression analysis was performed using a Generalized Linear Model (GLM) with a binomial family and logit link function, suitable for the binary outcome variable in our study—group membership of individuals as either HC or pMS. The Iteratively Reweighted Least Squares (IRLS) method was employed for its robustness in estimation when assumptions of normality might be violated. We chose this approach to ensure the most accurate and reliable interpretation of our complex dataset, which includes multiple covariates such as age, gender, EDSS score, and disease duration. We will enhance our manuscript by adding a subsection in the methods that further discusses the rationale behind our selection of these statistical methods.

  1. References are OK.

Thank you for your comment.

Round 2

Reviewer 1 Report

Comments and Suggestions for Authors

Ilona Součková et al. tried, for the most part, to answer the requirements of the review. However, in the last part of the Results section, there are still results that should be reformatted like this:

1) in lines 226-241, please express the p values ​​with a maximum of 4 decimal places (for situations when p= 1.49 * 10^-8 it will be written p<0.0001 according to the reports of the results in the medical research).

2) in tables 6 and 7, the statistics will be reported with 2-3 decimal places (please see Std.Error); in the case of confidence intervals with very small limits - it would be good to make a transformation of the variables (please see IL-1 ra, IL-6).

Author Response

Dear Madam, Sir,

Thank you for your valuable feedback. I understand your concern regarding the transformation of variables in Tables 6 and 7. Upon thorough review, it has been observed that the data in these tables already adhere to a normal distribution. The normality of the data was evaluated through violin plot (see fig. 3 and 4). Therefore, applying a transformation to these variables may not be necessary or appropriate, as it could potentially distort the natural variance and relationships within the data. However, I have ensured that the presentation of statistics, including the standard error, is precise and adheres to the standard practice of reporting with 2-3 decimal places.   Please let me know if there are any other aspects that I can clarify or modify. Best Regards, Michal Novotny
